# Quantum state tomography of molecules by ultrafast diffraction

Ming Zhang[1,8], Shuqiao Zhang[1,8], Yanwei Xiong [2], Hankai Zhang[1], Anatoly A. Ischenko [3], Oriol Vendrell[4], Xiaolong Dong[1], Xiangxu Mu[1], Martin Centurion [2], Haitan Xu[5,6 ✉], R. J. Dwayne Miller[7 ✉] & Zheng Li [1 ✉]

Ultrafast electron diffraction and time-resolved serial crystallography are the basis of the ongoing revolution in capturing at the atomic level of detail the structural dynamics of molecules. However, most experiments capture only the probability density of the nuclear wavepackets to determine the time-dependent molecular structures, while the full quantum state has not been accessed. Here, we introduce a framework for the preparation and ultrafast coherent diffraction from rotational wave packets of molecules, and we establish a new variant of quantum state tomography for ultrafast electron diffraction to characterize the molecular quantum states. The ability to reconstruct the density matrix, which encodes the amplitude and phase of the wavepacket, for molecules of arbitrary degrees of freedom, will enable the reconstruction of a quantum molecular movie from experimental x-ray or electron diffraction data.

[1] State Key Laboratory for Mesoscopic Physics and Collaborative Innovation Center of Quantum Matter, School of Physics, Peking University, Beijing 10087, China. [2] Department of Physics and Astronomy, University of Nebraska-Lincoln, Lincoln, NE, USA. [3] Lomonosov Institute of Fine Chemical Technologies, RTU-MIREA - Russian Technological University, Vernadskii Avenue 86, 119571 Moscow, Russia. [4] Physikalisch-Chemisches Institut, Universität Heidelberg, Im Neuenheimer Feld 229, D-69120 Heidelberg, Germany. [5] Shenzhen Institute for Quantum Science and Engineering, Southern University of Science and Technology, Shenzhen 518055, China. [6] School of Physical Sciences, University of Science and Technology of China, Hefei 230026, China. [7] Departments of Chemistry and Physics, University of Toronto, Toronto, ON M5S 3H6, Canada. [8]These authors contributed equally: Ming Zhang, Shuqiao Zhang. ✉email: xuht@sustech.edu.cn; dmiller@lphys.chem.utoronto.ca; zheng.li@pku.edu.cn

With the ability to directly obtain the Wigner function and density matrix of photon states, quantum tomography (QT) has made a significant impact on quantum optics[1–3], quantum computing[4,5] and quantum information[6,7]. By an appropriate sequence of measurements on the evolution of each degree of freedom (DOF), the full quantum state of the observed photonic system can be determined. The first proposal to extend the application of QT to reconstruction of complete quantum states of matter wavepackets[8] had generated enormous interest in ultrafast diffraction imaging[9–20] and pump-probe spectroscopy of molecules[21]. This interest was elevated with the advent of ultrafast electron and X-ray diffraction techniques using electron accelerators and X-ray free electron lasers to add temporal resolution to the observed nuclear and electron distributions[22,23]. In this respect, quantum tomography holds great promise to enable imaging of molecular wavefunctions beyond classical description. This concept could become a natural area for quantum tomography of quantum states of matter[24–28]. However, the great interest in this area has been tempered by the illustration of an "impossibility theorem", known as the dimension problem[29,30]. To obtain the density matrix of a system, the previoiusly established QT procedure relies on integral transforms (e.g. the tomographic Radon transform), which preserves dimensionality[1]. Unlike its quantum optics sibling, only a single evolutionary parameter, time, is available for the molecular wavepacket. This dimension problem prevents the use of existing QT algorithms to study quantum molecular dynamics because all rotational and most vibrational wavepackets cannot be retrieved.

Here we present an approach to resolve the notorious dimension problem. Solving this challenging problem is important to push imaging molecular dynamics to the quantum regime. Our approach makes quantum tomography a truly useful method in ultrafast physics which enables the making of quantum version of a "molecular movie"[12,17,27,28,31–34], without being limited in one dimension. As a proof-of-principle, we apply our QT retrieval approach to an ultrafast electron diffraction dataset to retrieve a quantum rotational wavepacket. We first demonstrate this approach using a numerical simulation of ultrafast diffraction imaging of laser-aligned nitrogen molecules[26]. The analysis with this QT approach correctly recovers the density matrix of the rotational wavepacket (schematically shown in Fig. 1), which is otherwise impossible to obtain with previously established QT procedures. We then apply this approach to ultrafast diffraction experiments to obtain the quantum density matrix from experimental data.

## Results

The modern formulation of quantum tomography based on integral transform[1,8,21] originates from the retrieval of wavefunction phases lost in the measurement. Dating back to 1933, Pauli and Feenberg proposed that a wavefunction $\psi(x, t) = |\psi(x, t)|e^{i\phi(x, t)}$ can be obtained by measuring the evolution of 1D position probability distribution $\Pr(x, t) = |\psi(x, t)|^2$ and its time derivative $\partial\Pr(x, t)/\partial t$ for a series of time points[35]. Equivalently, a pure quantum state can also be recovered by measuring $\Pr(x, t)$ at time $t$ and monitoring its evolution over short time intervals, i.e., $\Pr(x, t + N\Delta t) = |\psi(x, t + N\Delta t)|^2$ for $(N = 0, 1, 2, \cdots)$. Reconstructing the phase of wavefunction can be considered as the origin of quantum tomography. For a system with Hamiltonian $\hat{H} = \hat{H}_0 + \hat{H}_{\text{int}}$, the established 1D QT method makes use of knowledge of the non-interacting part of the Hamiltonian $\hat{H}_0$, so that its eigenfunctions can be precalculated and used in the tomographic reconstruction of density matrix through integral inversion transform. However, the dimension problem as demonstrated in the pioneering works[29,30] mathematically leads to singularity in the inversion from the

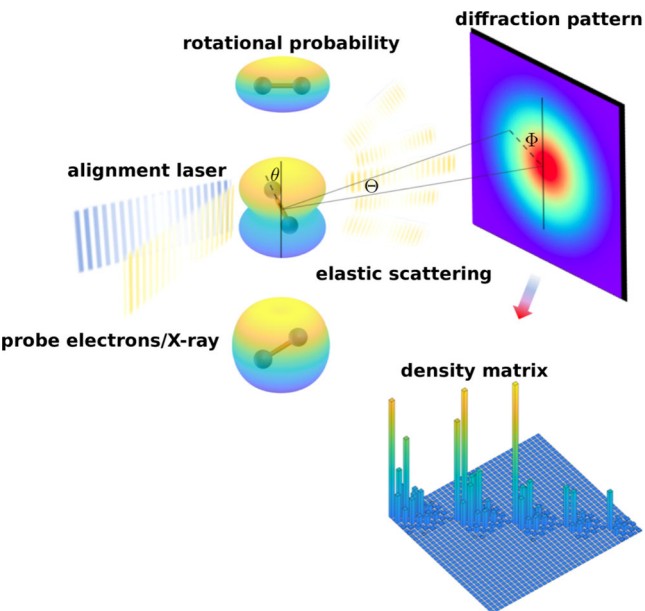

**Fig. 1 Schematic drawing of quantum tomography by ultrafast diffraction, illustrated with a rotational wavepacket of $N_2$ molecule.** A rotational wavepacket is prepared by an impulsive alignment laser pulse[42], and probed by diffraction of an incident electron/X-ray pulses for a series of time intervals. The mixed rotational quantum state represented by its density operator $\hat{\rho}$ is determined from the diffraction patterns.

evolving probability distribution to the density matrix and makes it challenging for higher dimensional QT.

We solve the QT dimension problem by exploiting the interaction Hamiltonian $\hat{H}_{\text{int}}$ and the analogy between QT and crystallographic phase retrieval (CPR)[36] in a seemingly distant field, crystallography. Further exploiting the interaction Hamiltonian $\hat{H}_{\text{int}}$ provides us a set of physical conditions, such as the selection rules of transitions subject to $\hat{H}_{\text{int}}$ and symmetry of the system. These physical conditions can be imposed as constraints in our QT approach, which is not feasible in the established QT methods based on integral transform. By compensating with the additional physical conditions as constraints in the iterative QT procedure, the converged solution can be obtained as the admissible density matrix that complies with all the intrinsic properties of the investigated physical system.

We start by presenting the correspondence between QT and CPR. The research on CPR has been the focus of crystallography for decades[9,24,34,36–38]. In crystallography, the scattered X-ray or electron wave encodes the structural information of molecules. The measured X-ray diffraction intensity is $I(\mathbf{s}) \sim |f(\mathbf{s})|^2$, where $\mathbf{s} = \mathbf{k}_f - \mathbf{k}_{\text{in}}$ is momentum transfer between incident and diffracted X-ray photon or electron, $f(\mathbf{s})$ is the electronically elastic molecular form factor. For X-ray diffraction, the form factor is connected to the electron density by a Fourier transform $f_X(\mathbf{s}) \sim \mathscr{F}[\Pr(\mathbf{x})]$, $\Pr(\mathbf{x})$ is the probability density of electrons in a molecule, and $\mathbf{x}$ is the electron coordinate. The form factor of electron diffraction has a similar expression $f_e(\mathbf{s}) = [\Sigma_\alpha N_\alpha \exp(i\mathbf{s} \cdot \mathbf{R}_\alpha) - f_X(\mathbf{s})]/s^2$, where $N_\alpha, \mathbf{R}_\alpha$ are the charge and position of $\alpha^{\text{th}}$ nucleus. However, the phase of the form factor, which is essential for reconstructing the molecular structure, is unknown in the diffraction experiment, only the modulus $|f(\mathbf{s})|$ can be obtained from measured diffraction intensity.

Phase retrieval is a powerful method that prevails in crystallography and single particle coherent diffraction imaging[24,37,38]. Its basic idea is illustrated in Fig. 2. Employing projective iterations between real space and Fourier space and imposing physical

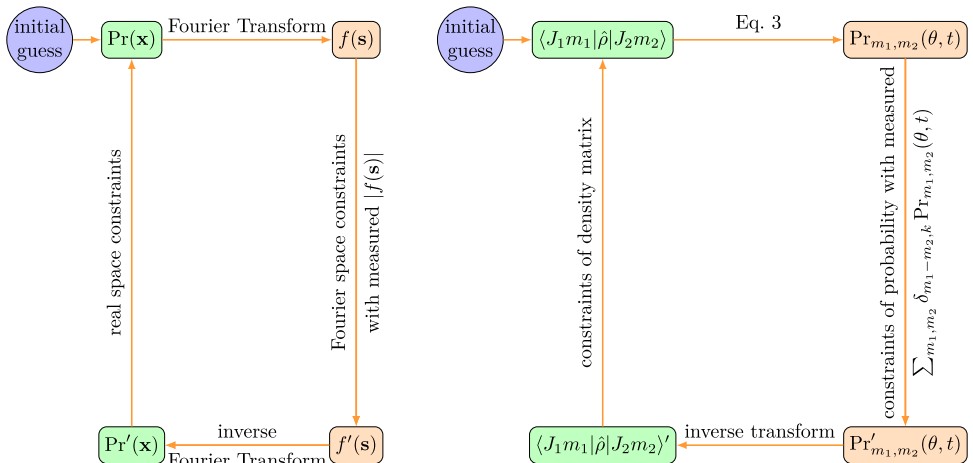

**Fig. 2 Analogy between crystallographic phase retrieval (CPR) and quantum tomography (QT) based on their common nature[35].** The CPR iteratively transforms between real space electron density $Pr(\mathbf{x})$ and Fourier space form factor $f(\mathbf{s})$ and impose constraints for both spaces, where Fourier space constraints comes from the measured diffraction intensity that provides the modulus of form factor $|f(\mathbf{s})|$. Analogously, QT iteratively transforms between blockwise probability distribution $Pr_{m_1,m_2}(\theta, t)$ in real space and elements in density matrix space, and the probability density evolution $Pr(\theta, \phi, t)$ is used to constraint the sum of blockwise probability distribution $Pr_{m_1,m_2}(\theta, t)$.

**Fig. 3 Quantum tomography of rotational wavepacket of nitrogen molecule.** The **a** moduli and **b** phases of density matrix elements. Within each $m$-block $J = |m|, |m| + 1, \cdots, J_{max}$ (phases are at $t = 0$). The density matrix elements of opposite magnetic quantum numbers $m$ and $-m$ are identical (see Supplementary Eq. 21). Density matrix elements of higher $m$-blocks are not plotted due to their small moduli. **c** The wavepacket probability distribution $Pr(\theta, t)$, which is cylindrically symmetric in azimuthal direction of $\phi$. The convergence of the procedure is illustrated in **d**, where the error of density matrix $\epsilon(\hat{\rho})$ and the error of probability density $\epsilon(Pr)$ are defined in Supplementary Eq. 33 and 34.

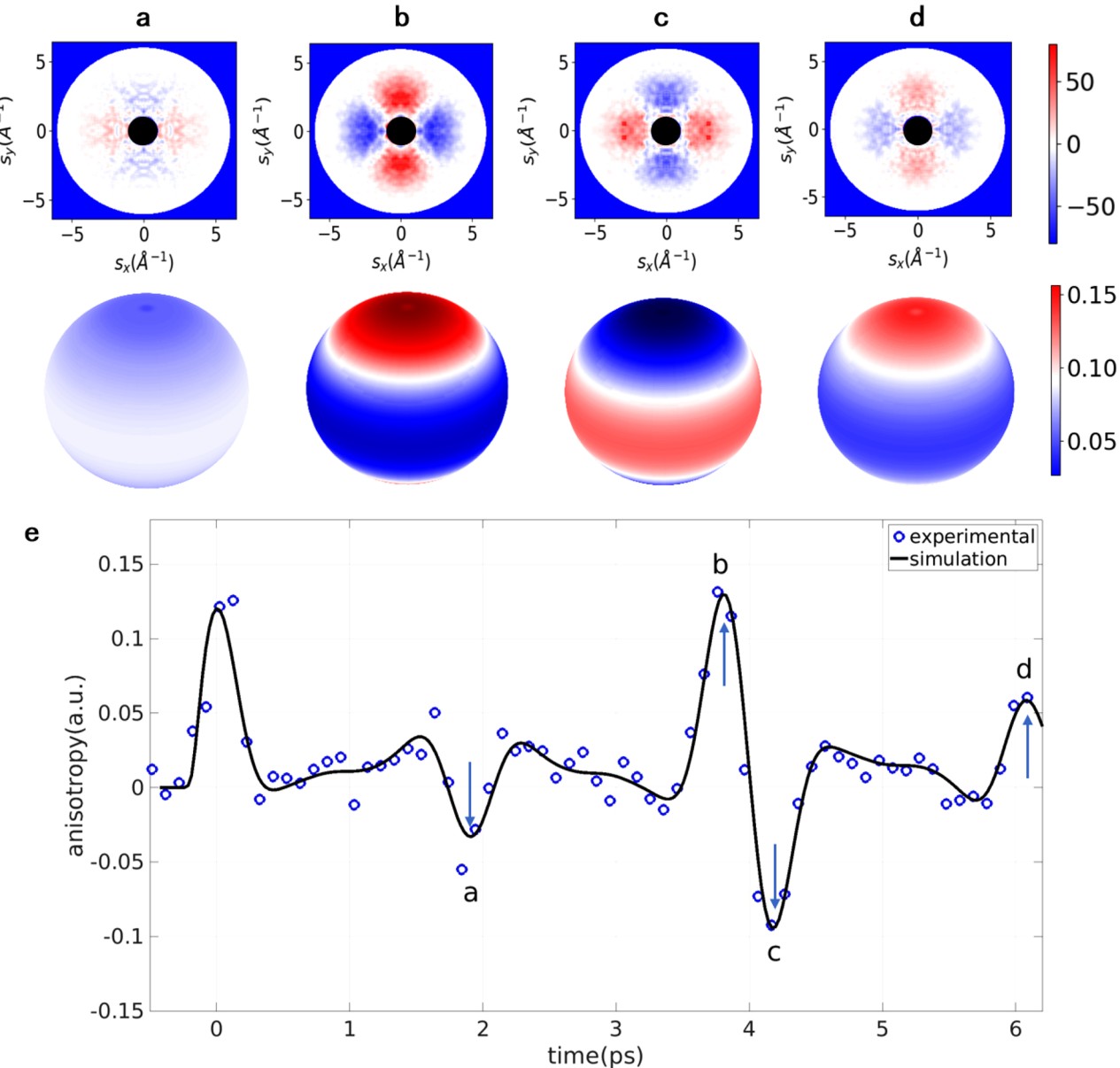

**Fig. 4 Experimental ultrafast electron diffraction data for N$_2$ rotational wavepacket.** Difference-diffraction pattern and the angular probability distribution Pr$(\theta, \phi, t)$ at various delay times marked in **e**: **a** $t = 1.9$ ps, **b** $t = 3.8$ ps, **c** $t = 4.2$ ps, **d** $t = 6.1$ ps. The dark circle corresponds to the regions where scattered electrons are blocked by the beam stop. **e** Temporal evolution of the experimental and simulated anisotropy of the rotational wavepacket.

constraints in both spaces, the lost phases of the form factor $f(\mathbf{s})$ can be reconstructed with high fidelity. Fourier space constraint utilizes measured diffraction intensity data, and real space constraints comes from a priori knowledge, e.g. the positivity of electron density. We present the new method of quantum tomography based on this conceptual approach by applying it to rotational wavepackets of nitrogen molecules prepared by impulsive laser alignment, using the ultrafast electron diffraction (UED). Quantum tomography of rotational wavepackets is impossible in the previously established QT theory, because the full quantum state of a rotating linear molecule is a 4D object $\langle\theta, \phi|\hat{\rho}|\theta', \phi'\rangle$, while the probability density evolution Pr$(\theta, \phi, t)$ extracted from measured diffraction patterns is only 3D. It is obvious that the inversion problem to obtain the density matrix is not solvable by dimensionality-preserving transform.

We first demonstrate the capability of our approach to correctly recover the density matrix despite the dimension problem,

using numerical simulation of ultrafast diffraction of impulsively aligned nitrogen molecule with an arbitrarily chosen temperature of 30 K. The order of recovered density matrix sets the requirement on the resolution. From Eq. (3), the characteristic time scale of rotation is $\frac{1}{\Delta\omega} = \frac{2\mathcal{I}}{|\Delta J|(J+1)}$, where $\mathcal{I}$ is the moment of inertia of nitrogen molecule, $\Delta J = J_1 - J_2$ and $J = J_1 + J_2$ for any two eigenstates with $J_1, J_2$. Using the Nyquist-Shannon sampling theorem, the required temporal resolution $\delta t$ should be $\delta t \leq \frac{1}{2\Delta\omega}$. The spatial resolution $\delta\theta$ and $\delta\phi$ can be determined with the argument that the nodal structure of spherical harmonic basis in Eq. (2) must be resolved, i.e. $\delta\theta < \frac{\pi}{2J_{\max}}$. To recover density matrix up to the order $J_{\max} = 8$, it demands time resolution $\delta t \sim 10^2$ fs and spatial resolution $\delta\theta \sim 10^{-1}$ rad. Quantum tomography of the rotational wavepacket gives the result shown in Fig. 3. After 50 iterations, both density matrix and probability distribution are precisely recovered. The error of density matrix is $\epsilon_{50}(\hat{\rho}) = 2.9 \times 10^{-2}$ and error of probability achieves $\epsilon_{50}(\text{Pr}) = 3.8 \times 10^{-5}$

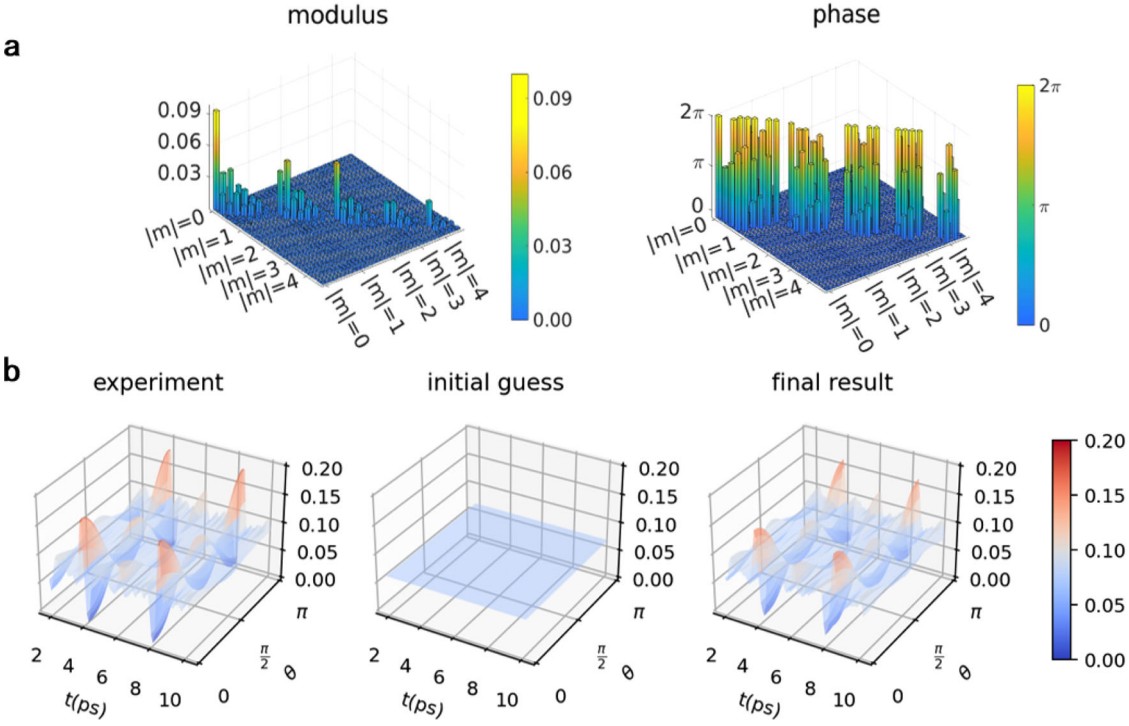

**Fig. 5 Experimental quantum tomography of rotational wavepacket of nitrogen molecule. a** The moduli and phases of QT retrieved density matrix elements. Within each $m$-block $J = |m|, |m| + 1, \cdots, J_{max}$ (phases are plotted at $t = 1.95$ ps after the alignment pulse). The density matrix elements of opposite magnetic quantum numbers $m$ and $-m$ are identical (see Supplementary Eq. 21). Density matrix elements of higher $m$-blocks are not plotted due to their small moduli. **b** The wavepacket probability distribution $\Pr(\theta, t)$ (cylindrically symmetric in azimuthal direction of $\phi$) of experimental data, initial guess and final result of QT.

(see Supplementary Eqs. 33 and 34 for the definition of $\epsilon(\hat{\rho})$ and $\epsilon(\Pr)$).

We then apply this iterative QT method to the ultrafast electron diffraction (UED) experiment to extract the quantum density matrix of $N_2$ rotational wavepacket, prepared at a temperature of 45 K. The experimental parameters are described in detail in a previous publication[39]. We use a tabletop kilo-electron-volt (keV) gas-phase UED setup to record the diffraction patterns of nitrogen molecules that are impulsively aligned by a femtosecond laser pulse. The details of the keV UED setup has been introduced in[39,40], which is schematically shown in Fig. 1. Briefly, an 800 nm pump laser pulse with a pulse duration of 60 fs (FWHM) and pulse energy of 1 mJ is used to align the molecules. A probe electron pulse with kinetic energy of 90 keV and 10,000 electrons per pulse is used and the diffraction pattern of the electrons scattered from the molecules is recorded. The nitrogen molecules are introduced in a gas jet using a de Laval nozzle. The laser pulse has a tilted pulse front to compensate the group velocity mismatch between the laser and electron pulses, and an optical stage is used to control the time delay between the pump and probe pulse with a time step of 100 fs. The pump laser launches a rotational wave packet, which exhibits dephasing and subsequent revivals of alignment in picosecond time scale. The experimental diffraction patterns at several time delays are shown in Fig. 4a–d. The temporal evolution of diffraction patterns can be characterized by the anisotropy, defined as $(S_H - S_V)/(S_H + S_V)$, where $S_H$ and $S_V$ are the sum of the counts in horizontal and vertical cones in the diffraction patterns at $3.0 < s < 4.5$ Å$^{-1}$, with an opening angle of 60 degrees. The temporal evolution of angular probability distribution $\Pr(\theta, \phi, t)$ can be retrieved using the method described in[39], followed by a deconvolution using a point spread function with FWHM width of 280 fs to remove the

blurring effect due to the limited temporal resolution of the setup. Data is recorded from before excitation of the laser up to 6.1 ps after excitation. In order to complete the data up to a full cycle, which is needed for the quantum tomography, the angular probability distribution evolution is extended to obtain the data from 6.1 ps to 11 ps using a reflection of the data from 6.1 ps to 1.2 ps based on the symmetry of the evolution of the rotational wavepacket. The diffraction patterns and corresponding angular distributions at various time delays are shown in Fig. 4. Using our QT method, we obtain the complex density matrix in Fig. 5, which completely determines the rotational quantum state of the system. The error of recovered probability distribution converges to $\epsilon(\Pr) = 6.4 \times 10^{-2}$. The difference between recovered angular probability distribution and the experimental result comes from the restriction of order of recovered density matrix due to limited temporal and angular resolution in the experiment.

## Discussion

In summary, we have demonstrated an iterative quantum tomography approach that is capable of extracting the density matrix of high-dimensional wavepacket of molecules from its evolutionary probability distribution in time. The notorious dimension problem, which has prohibited for almost two decades the quantum tomographic reconstruction of molecular quantum state from ultrafast diffraction, has thus been resolved. This quantum tomography approach can be straightforwardly extended to obtain other quantum states, such as vibrational states (see Supplementary Note 8) or electronic states. The retrieval of the full density matrix can be used to study important new information about the quantum dynamics. For example, the passage of a nuclear wavepacket through a conical intersection, a region in the potential energy surface where the ground excited states

intersect, has been widely studied[41] because it is crucial for the relaxation of photoexcited molecules. A conical intersection produces a coherent superposition of two states which can be directly observed in the off-diagonal terms of the density matrix. Thus, a full quantum retrieval would provide direct evidence of the presence of a conical intersection and the moment at which the wavepacket reaches it. We expect this advance to have a broad impact in many areas of science and technology, not only for making the quantum version of molecular movies, but also for QT of other systems when quantum state information is tainted by insufcient evolutionary dimensions or incomplete measurements.

## Methods

**Iterative quantum tomography**. From a dataset consisting of a series of time-ordered snapshots of diffraction patterns

$$I(\mathbf{s}, t) = \int_0^{2\pi} d\phi \int_0^{\pi} \sin\theta d\theta \Pr(\theta, \phi, t)|f(\mathbf{s}, \theta, \phi)|^2, \tag{1}$$

where the form factor $f$ is related to the molecule orientation. The time-dependent molecular probability distribution $\Pr(\theta, \phi, t)$ can be obtained by inverse Fourier and Abel transform[39] or by solving the Fredholm integral equation of the first kind (see Supplementary Note 6 for details). The probability distribution of a rotational wavepacket is

$$\Pr(\theta, \phi, t) = \sum_{J_1 m_1} \sum_{J_2 m_2} \langle J_1 m_1 | \hat{\rho} | J_2 m_2 \rangle Y_{J_1 m_1}(\theta, \phi) Y_{J_2 m_2}^*(\theta, \phi) e^{-i\Delta\omega t}, \tag{2}$$

where $\Delta\omega = \omega_{J_1} - \omega_{J_2}$ is the energy spacing of rotational levels. As shown in Fig. 2, we devise an iterative procedure to connect the spaces of density matrix and temporal wavepacket density. For the system of rotational molecules, the dimension problem limits the invertible mapping between density matrix and temporal wavepacket density to the reduced density of fixed projection quantum numbers $m_1$, $m_2$,

$$\Pr_{m_1, m_2}(\theta, t) = \sum_{J_1, J_2} \langle J_1 m_1 | \hat{\rho} | J_2 m_2 \rangle \tilde{P}_{J_1}^{m_1}(\cos\theta) \tilde{P}_{J_2}^{m_2}(\cos\theta) e^{-i\Delta\omega t}, \tag{3}$$

where $\tilde{P}_J^m(\cos\theta)$ is the normalized associated Legendre polynomial defined in Supplementary Eq. 2. The analytical solution of the inverse mapping from $\Pr_{m_1, m_2}(\theta, t)$ to density matrix $\langle J_1 m_1 | \hat{\rho} | J_2 m_2 \rangle$ is elaborated in Supplementary Note 3. However, due to the dimension problem, there is no direct way to obtain $\Pr_{m_1, m_2}(\theta, t)$ from the measured wavepacket density, only their sum is traceable through $\sum_{m_1, m_2} \delta_{m_1 - m_2, k} \Pr_{m_1, m_2}(\theta, t) = \int_0^{2\pi} \Pr(\theta, \phi, t) e^{ik\phi} d\phi$.

Our method starts from an initial guess of density matrix and an iterative projection algorithm is used to impose constraints in the spaces of density matrix and spatial probability density. The initial guess of quantum state, $\hat{\rho}_{ini} = \sum_{J_0 m_0} \omega_{J_0} | J_0 m_0 \rangle \langle J_0 m_0 |$, is assumed to be an incoherent state in the thermal equilibrium of a given rotational temperature, which can be experimentally determined[26]. $\omega_{J_0} = \frac{1}{Z} g_{J_0} e^{-\beta E_{J_0}}$ is the Boltzmann weight, and $g_{J_0}$ represents the statistical weight of nuclear spin, for the bosonic $^{14}N_2$ molecule, $g_{J_0}$ is 6 for even $J_0$ (spin singlet and quintet) and 3 for odd $J_0$ (spin triplet).

In the probability density space, constraint is imposed by uniformly scaling each reduced density $\Pr_{m_1, m_2}(\theta, t)$ with the measured total density $\Pr(\theta, \phi, t)$. Constraints in the density matrix space enable us to add all known properties of a physical state to the QT procedure, which supply additional information to compensate the missing evolutionary dimensions. The constraints contain general knowledge of the density matrix, i.e. the density matrix is positive semidefinite, Hermitian and with a unity trace. Besides, the selection rules of the alignment laser-molecule interaction imply further constraints on physically nonzero $m$-blocks of the density matrix and invariant partial traces of density matrix elements subject to projection quantum number $m$ (see Supplementary Note 5 for details of the algorithm).

## Data availability
The data that support the plots within this paper and other findings of this study are available from the corresponding authors upon reasonable request.

## Code availability
The codes for quantum tomography algorithm are available from the corresponding authors upon reasonable request.

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

## Acknowledgements

We thank Jie Yang, Yi-Jen Chen, Zunqi Li and Stefan Pabst for useful discussions. This work was supported by NSFC Grant No. 11974031, funding from state key laboratory of mesoscopic physics, RFBR Grant No. 20-02-00146, and NSERC. Y.X. and M.C. were supported by the National Science Foundation, Physics Division, Atomic, Molecular and Optical Sciences program, under Award No. PHY-1606619.

## Author contributions

Z.L. designed the study. M.Z., S.Q.Z., H.K.Z. and Z.L. carried out the calculations. M.Z., S.Q.Z., H.K.Z., X.L.D., X.X.M, H.X., O.V., R.J.D.M., A.I. and Z.L. analyzed the data. Y.X., M.C., and M.Z. analyzed the experimental data. All authors contributed to the writing of the manuscript.

## Competing interests

The authors declare no competing interests.
