## [Peer Review File · Nature Communications]

REVIEWER COMMENTS

Reviewer #1 (Remarks to the Author):

Review

Quantum state tomography of molecules by ultrafast diffraction

Nature Comm

Zhang et al

Zhang et al theoretically and experimentally present the method of quantum tomography to reconstruct the density matrix of molecules. They make use of an analogy with crystallographic phase retrieval (CPR) that is often used in the structure determination of materials/molecules with X-ray/electron diffraction tools. By making use of constraints in the iterative retrieval procedure (similar as in CPR), they are able to overcome the notorious dimensionality problem in quantum tomography. They demonstrate their approach on simulated and real data on a laser-aligned rotating linear molecule (dinitrogen), for which they are able to obtain the full rotational density matrix. Their method can now be extended to larger molecules (more degrees of freedom).

While the author's approach seems of great importance, I think the paper is hard to follow and quite specialized. I have a decent background in quantum mechanics and crystallography, yet I had to read the manuscript many times to make sense of it. I also wonder whether this is a true breakthrough in quantum tomography, or whether this is "just" an elegant combination of two fields. I suggest that the authors make the article more accessible to a general audience (such as expected for Nature Comm) by removing jargon and stress more how this is a breakthrough in quantum tomography/science.

A more specific comment: on p. 6 the authors write "In order to complete the data up to a full cycle, which is needed for the quantum tomography, the angular probability distribution evolution is extended to obtain the data from 6.1 ps to 11 ps using a reflection of the data from 6.1 ps to 1.2 ps based on the symmetry of the evolution of the rotational wavepacket." I don't see this symmetry in the data. How can you know the period? Could the authors measure a full period instead?

Just above that sentence the authors say they deconvoluted the instrumental response function. They should show (in the SI) how they determined the latter. ^[L]_{SEP}

Reviewer #2 (Remarks to the Author):

The present work demonstrates important advances in quantum tomography, i.e., the reconstruction of the quantum state (density matrix) from experimental measurements.

It is demonstrated both using a numerical simulation of ultrafast diffraction data as well as from experimental data, that the procedure recovers the density matrix of the rotational wavepacket of laser-aligned nitrogen molecules.

There is one point that deserves further justification: The work is placed in the context of ultrafast structural dynamics of molecules, i.e., “molecular movies”. In the context of a full quantum version of a “molecular movie”, it seems to me that the desired information is the temporal evolution of the (quantum) probability density? Please explain why the density matrix is of interest for “molecular movies”.

I have a couple of more specific comments to the manuscript:

Line 103: The wording “...measured probability density...”, the diffraction patterns are measured and the probability density is extracted from these measurements. To that end, in line 107: “...probability distribution ... can be obtained by solving the Fredholm integral equation...”. This procedure is described in the supplementary information (SI, page 16). However, an alternative procedure is described on page 10 of the SI, which procedure is actually used?

Concerning the experimental data: What is temporal duration of probe pulse – including timing jitter? I suppose that the “deconvolution using a point spread function with FWHM width of 280 fs” is related to this question. What is the spatial resolution that can be obtained with the 90 keV electron pulses? To that end, comment on the current status of X-ray free electron lasers.

Reviewer #3 (Remarks to the Author):

In their manuscript Zhang and co-workers present the quantum state tomography of rotational states from time evolving gas-phase diffraction images. They demonstrate their method by reconstructing the density matrix of rotational states of the nitrogen molecule measured by ultra fast electron diffraction. The sample is initially aligned by a strong alignment laser pulse. I think this an interesting idea which addresses the central problem of crystallography (namely phase retrieval) from a different perspective. As such the paper is of interest for a broader community. The paper is well written and publishable.

The authors may want to comment or give an outlook on how their scheme could be extended beyond rotational states. Quantum states tomography of vibrational or electronic states will be application that the community will look forward to.

Zhang et al theoretically and experimentally present the method of quantum tomography to reconstruct the density matrix of molecules. They make use of an analogy with crystallographic phase retrieval (CPR) that is often used in the structure determination of materials/molecules with X-ray/electron diffraction tools. By making use of constraints in the iterative retrieval procedure (similar as in CPR), they are able to overcome the notorious dimensionality problem in quantum tomography. They demonstrate their approach on simulated and real data on a laser-aligned rotating linear molecule (dinitrogen), for which they are able to obtain the full rotational density matrix. Their method can now be extended to larger molecules (more degrees of freedom).

Reply: We are grateful to the Referee for the positive evaluation of our work and helpful suggestions.

While the author's approach seems of great importance, I think the paper is hard to follow and quite specialized. I have a decent background in quantum mechanics and crystallography, yet I had to read the manuscript many times to make sense of it. I also wonder whether this is a true breakthrough in quantum tomography, or whether this is "just" an elegant combination of two fields. I suggest that the authors make the article more accessible to a general audience (such as expected for Nature Comm) by removing jargon and stress more how this is a breakthrough in quantum tomography/science.

Reply: We have revised the manuscript to make the manuscript more accessible for a general audience and more clearly explained the significance of this work. We believe this is a true breakthrough because it demonstrates theoretically and experimentally the retrieval of the full quantum state of the rotational wavepacket of molecules rather than just the probability density. This has been an outstanding problem for many years. The density matrix encodes the complete information of the system and thus carries more information than the probability density. Therefore, it generalizes the concept of the "molecular movie", which used to refer only to the probability density, to the full quantum mechanical state. This is illustrated with a proof-of-principle experiment of ultrafast electron diffraction to retrieve the quantum rotational wavepackets. Our approach can, in principle, be applied to obtain quantum states in other experiments as well. The retrieval of the full density matrix can be used to retrieve important new information about the quantum dynamics, e.g., the study of the relaxation of photoexcited molecules. We have emphasized the importance of retrieving the full density matrix, and the potential of this method to have a large impact on the field. Undoubtedly other applications will come up where the information about the coherence or phase of the wavepacket is relevant, e.g. in the

main text of the revised manuscript, we have modified the abstract and added perspective on application of quantum tomography in resolving the conical intersection of non-adiabatic dynamics.

A more specific comment: on p. 6 the authors write “In order to complete the data up to a full cycle, which is needed for the quantum tomography, the angular probability distribution evolution is extended to obtain the data from 6.1 ps to 11 ps using a reflection of the data from 6.1 ps to 1.2 ps based on the symmetry of the evolution of the rotational wavepacket.” I don’t see this symmetry in the data. How can you know the period? Could the authors measure a full period instead?

Reply: The period of the motion is $T = h/2B$, where B is the rotational constant of the molecule (which depends on the moment of inertia), which is easy to determine since the bond length of the nitrogen molecule is well known. The motion is periodic because the molecule is linear so there is only one relevant rotational constant. It is a good point that it would have been better to measure a full period experimentally, however, the experiment is very time consuming and the timing stability of the setup cannot be maintained typically beyond one overnight run. One could take data over several days but we wanted to avoid stitching the data together as this might introduce artifacts. Figure R1 below shows why we are confident that this method does not introduce artifacts. The figure shows the measured and the extrapolated data, along with the simulated signal and the simulated signal extrapolated from the first half-period. While the simulation and the extrapolated simulated signal are not identical, the difference between the two is much smaller than the uncertainty in the experimental measurement. To summarize, we agree with the referee that it would be better to have complete data, but in this case it is not feasible. However, we are confident that this extrapolation does not introduce any artifacts.

Figure R1. Temporal evolution of the nitrogen alignment anisotropy. The blue circles show the experimental anisotropy; the solid red line is the simulation; the blue triangles for $t > 6.1$ ps show the extended anisotropy which are obtained by using the anisotropy from 6.1 ps to 1.2 ps; the green circles for $t > 6.1$ ps show the reflection of the simulated anisotropy from 6.1 ps to 1.2 ps.

Just above that sentence the authors say they deconvoluted the instrumental response function. They should show (in the SI) how they determined the latter.

Reply: The instrumental response function of the UED setup is assumed to be a Gaussian function with the full width half maximum corresponding to the temporal resolution of the setup. We compare the deconvoluted angular distribution evolution with the simulation to obtain the general temporal resolution, which is 280 fs in the manuscript. We have added the information about how the instrument response function was determined in the SI.

Report of the Second Referee --- NCOMMS-21-15820

The present work demonstrates important advances in quantum tomography, i.e., the reconstruction of the quantum state (density matrix) from experimental measurements.

Reply: We thank the Referee for the positive evaluation of our work and helpful suggestions.

It is demonstrated both using a numerical simulation of ultrafast diffraction data as well as from experimental data, that the procedure recovers the density matrix of the rotational wavepacket of laser-aligned nitrogen molecules.

There is one point that deserves further justification: The work is placed in the context of ultrafast structural dynamics of molecules, i.e., “molecular movies”. In the context of a full quantum version of a “molecular movie”, it seems to me that the desired information is the temporal evolution of the (quantum) probability density? Please explain why the density matrix is of interest for “molecular movies”.

Reply: The density matrix encodes all the physical information of quantum state and therefore represents a generalization of the concept of “molecular movie” beyond the classical “ball-and-stick” picture. The temporal evolution of the probability density distribution, the usual “molecular movie”, is only one of the observables that can be derived from the density matrix. We have shown the retrieval of the density matrix at different time instant ($t=0$ in Fig. 3 and $t=1.95$ ps in Fig. 5 of the main text) as a proof of principle. In addition, we have added a discussion in the revised main text about a wavepacket passage through conical intersections, where the information of the full

density matrix would also provide unique information, which cannot be accessed by the probability density alone.

I have a couple of more specific comments to the manuscript:

Line 103: The wording "...measured probability density...", the diffraction patterns are measured and the probability density is extracted from these measurements. To that end, in line 107: "...probability distribution ... can be obtained by solving the Fredholm integral equation...". This procedure is described in the supplementary information (SI, page 16). However, an alternative procedure is described on page 10 of the SI, which procedure is actually used?

Reply: We used the procedure described on page 10 of the SI. The two procedures presented in the SI are both based on Eq. (1) in the main text, and can extract the probability density from the rotationally averaged diffraction pattern. The inverse Fourier and Abel transformation method is actually used when retrieving N₂ rotational wavepacket from experimental data. We also provided an alternative method by solving the Fredholm integral equation. For clarity, we revised the text below Eq. (1), and provided two procedures, which are equivalent in principle for the extraction of the probability density from diffraction pattern.

Concerning the experimental data: What is temporal duration of probe pulse – including timing jitter? I suppose that the "deconvolution using a point spread function with FWHM width of 280 fs" is related to this question. What is the spatial resolution that can be obtained with the 90 keV electron pulses? To that end, comment on the current status of X-ray free electron lasers.

Reply: The temporal duration of the probe pulse is estimated to be around 200 fs. 280 fs is the overall temporal resolution of the setup, including pump pulse duration, probe pulse duration, temporal broadening due to group velocity mismatch and timing jitter. The spatial resolution, meaning the width of a peak in the pair distribution function, is typically between 0.5 and 0.8 Angstroms and is only limited by the maximum momentum transfer captured in the diffraction pattern (typically between 7 and 12 Angstrom⁻¹). An interatomic distance can be determined with a precision of approximately 0.05 Angstroms, depending on the signal levels of specific experiments. X-ray diffraction using XFELs usually captures a momentum transfer of around 5 Angstrom⁻¹.

Report of the Third Referee --- NCOMMS-21-15820

In their manuscript Zhang and co-workers present the quantum state tomography of rotational states from time evolving gas-phase diffraction images. They demonstrate their method by reconstructing the density matrix of rotational states

of the nitrogen molecule measured by ultra fast electron diffraction. The sample is initially aligned by a strong alignment laser pulse. I think this an interesting idea which addresses the central problem of crystallography (namely phase retrieval) from a different perspective. As such the paper is of interest for a broader community. The paper is well written and publishable.

Reply: We are thankful to the Referee for the positive evaluation of our work and helpful suggestions.

The authors may want to comment or give an outlook on how their scheme could be extended beyond rotational states. Quantum states tomography of vibrational or electronic states will be application that the community will look forward to.

Reply: We thank the Referee for the suggestion. We have added discussion in the manuscript about the applications of QT of quantum states beyond rotational states, and a detailed demonstration of quantum tomography of vibrational quantum states and the overcoming of dimension problem in the last section of Supplementary Information, named as “Vibrational Quantum Tomography”.

REVIEWERS' COMMENTS

Reviewer #1 (Remarks to the Author):

All concerns have been satisfactorily addressed. The manuscript can be accepted for publication in Nat Comm.

Reviewer #2 (Remarks to the Author):

I find that my remarks to the first version of the manuscript have been adequately addressed in the revised manuscript.

Reviewer #3 (Remarks to the Author):

The authors have now added a section addressing the reconstruction of vibrational density matrix, which is very interesting. I can recommend publication in Nat. Comm.

I have one question left here: it appears that the reconstruction of the vibrational density is based on a harmonic model. How would a strong anharmonicity in a vibrational mode affect the reconstruction?

Report of the First Referee --- NCOMMS-21-15820

All concerns have been satisfactorily addressed. The manuscript can be accepted for publication in Nat Comm.

Reply: We thank the Referee for the positive evaluation of our work.

Report of the Second Referee --- NCOMMS-21-15820

I find that my remarks to the first version of the manuscript have been adequately addressed in the revised manuscript.

Reply: We thank the Referee for the positive evaluation of our work.

Report of the Third Referee --- NCOMMS-21-15820

The authors have now added a section addressing the reconstruction of vibrational density matrix, which is very interesting. I can recommend publication in Nat. Comm.

I have one question left here: it appears that the reconstruction of the vibrational density is based on a harmonic model. How would a strong anharmonicity in a vibrational mode affect the reconstruction?

Reply: We are thankful to the Referee for the positive evaluation of our work and helpful suggestions. A strong anharmonicity in a vibrational mode does not affect the reconstruction. In Supplementary Information, we established the quantum tomography procedure of multiple vibrational modes with strong coupling. The wavefunction ansatz and the self-consistent field state interaction (SCF-SI) method can be extended to the treatment of molecular vibrations with anharmonic potential and strong coupling (see J. M. Bowman et al., J. Phys. Chem. 83, 905 (1979)). To solve the vibrational Hamiltonian efficiently for a general case, the separable part for the anharmonic vibrational modes of Hamiltonian in Supplementary Eq. 45 can be replaced by, e.g., Morse potential model. The transformation from probability distribution subspace to density matrix space (Supplementary Eq. 48) can also be established using the sampling function of Morse potential (see U. Leonhardt, Phys. Rev. A. 55, 3164 (1997)). In the revised Supplementary Information, we have added the discussion above in Supplementary Note 8 as follows: “The wavefunction ansatz and the self-consistent field state interaction (SCF-SI) method can be extended to the treatment of molecular vibrations with anharmonic potential and strong coupling. To

solve the vibrational Hamiltonian efficiently for this general case, the separable part Hamiltonian \hat{h}_i for the anharmonic vibrational modes can be replaced by, for example, Morse potential model and corresponding pattern function can be used.”